# Identification of Prognosis-Related Oxidative Stress Model with Immunosuppression in HCC

**DOI:** 10.3390/biomedicines11030695

**Published:** 2023-02-24

**Authors:** Zhixuan Ren, Jiakang Zhang, Dayong Zheng, Yue Luo, Zhenghui Song, Fengsheng Chen, Aimin Li, Xinhui Liu

**Affiliations:** 1Integrated Hospital of Traditional Chinese Medicine, Southern Medical University, Guangzhou 510315, China; 2Cancer Center, Southern Medical University, Guangzhou 510315, China

**Keywords:** oxidative stress, hepatocellular carcinoma, prognostic model, immune microenvironment, immunotherapy

## Abstract

For hepatocellular carcinoma (HCC) patients, we attempted to establish a new oxidative stress (OS)-related prognostic model for predicting prognosis, exploring immune microenvironment, and predicting the immunotherapy response. Significantly differently expressed oxidative stress-related genes (DEOSGs) between normal and HCC samples from the Cancer Genome Atlas (TCGA) were screened, and then based on weighted gene coexpression network analysis (WGCNA), HCC-related hub genes were discovered. Based on the least absolute shrinkage and selection operator (LASSO) and cox regression analysis, a prognostic model was developed. We validated the prognostic model’s predictive power using an external validation cohort: the International Cancer Genome Consortium (ICGC).Then a nomogram was determined. Furthermore, we also examined the relationship of the risk model and clinical characteristics as well as immune microenvironment. 434 DEOSGs, comprising 62 downregulated and 372 upregulated genes (*p* < 0.05 and |log2FC| ≥ 1), and 257 HCC-related hub genes were recognized in HCC. Afterward, we built a five-DEOSG (*LOX*, *CYP2C9*, *EIF2B4*, *EZH2*, and *SRXN1*) prognostic risk model. Using the nomogram, the risk model was shown to have good prognostic value. Compared to the low risk group, HCC patients with high risk had poorer outcomes, worse pathological grades, and advanced tumor stages (*p* < 0.05). There were significant increases in *LOX*, *EIF2B4*, *EZH2*, and *SRXN1* expression in HCC samples, while *CYP2C9* expression was decreased. Finally, Real-time PCR (RT-qPCR) confirmed the mRNA expressions of five genes (*CYP2C9*, *EIF2B4*, *EZH2*, *SRXN1*, *LOX*) in HCC cell lines. Our study constructed a prognostic OS-related model with strong predictive power and potential as an immunosuppressive biomarker for HCC leading to improving prediction and providing new insights for HCC immunotherapy.

## 1. Introduction

Globally, primary liver cancer is the sixth-most common malignant tumor with ∼900,000 newly diagnosed patients annually, and is the third-most major cause of cancer-related mortality [1]. Approximately 75% to 85% of primary liver cancer cases are HCC, which are difficult to diagnose early and have a poor long-term prognosis [2]. For early HCC patients, radical surgery and transplantation are still the main radical therapies. The fact remains that a large number of HCC patients do not receive a diagnosis until they are in advanced stages and thus miss out on curative treatment due to the lack of apparent symptoms in the early stages [3,4]. Targeted molecular therapies, including sorafenib, lenvatinib, regorafenib, and immunotherapy have shown promising results in treating HCC [5,6,7]. Despite this, HCC patients continue to have dismal clinical outcomes in consequence of the emerged resistance. Consequently, many studies were performed to explore more effective models to precisely predict the prognosis and response to treatment of HCC patients for providing more evidence for precision therapy. Regardless of these explorations, rare models have been used in the clinic application of HCC patients. Consequently, finding new validated prognostic biomarkers and developing new models to predict HCC patients’ prognosis and treatment are crucial.

Oxidative stress (OS) occurs from the overproduction of reactive oxygen species (ROS) and reactive nitrogen species (RNS) [8], and is associated with the generation and progression of various physiological events or diseases, such as cancer, obesity and diabetes [9,10]. It has been found that increased ROS and RNS stimulate cell proliferation and angiogenesis in cancer cells and contribute to cancer progression [11]. ROS and RNS production are increased by hypoxia-inducible factors (HIFs) through upregulation of NADPH oxidases (NOXs) and nitric oxide synthases (NOSs) by binding to the hypoxic response element (HRE) in their promoter regions, respectively [12]. Accumulating evidence suggests that ROS, including superoxide free radicals, nitric oxide free radicals, hydroxyl free radicals, and uncharged substances, are significantly increased in HCC [13,14]. Recent studies have shown that the contribution of ROS to HCC is a complicated and varied process that involves their interacting directly with proteins and regulating gene expression or transcription factors to regulate multiple signal transduction pathways, while the overproduction of RNS causes protein damage in HCC [15,16]. Intracellular ROS accumulation can significantly promote tumor formation and growth, and induce drug resistance in HCC [13]. Moreover, several therapeutic approaches targeting OS have been proposed as possible therapies for HCC [15]. However, to our best knowledge, whether OS-related genes can accurately predict HCC prognosis and immunotherapy response has not yet been systematically evaluated, and the underlying mechanisms need to be further investigated.

The tumor microenvironment plays an important role in the development of tumors, especially in HCC [17]. As a typical inflammation driven tumor, HCC can promote immune tolerance through its immunosuppressive microenvironment. The dynamic balance of oxidative stresses not only orchestrate complex cell signaling events in cancer cells, but also affect other components in the tumor microenvironment (TME) [18]. Immune cells, such as M2 macrophages, dendritic cells, and T cells, are the major components of the immunosuppressive TME from the ROS-induced inflammation. As cancer treatment advances, immunotherapies are emerging as the next frontier, and HCC’s immunobiology is an area that will need further research. Immune checkpoint molecules contribute to HCC immunosuppressive through suppressing the anti-tumor immune response [19]. Immune checkpoint inhibitor (ICI) is one of the most rapidly developed immunotherapy strategies of HCC in recent years. ICI can block tumor-induced immunosuppression, thereby enhancing the anti-tumor immune response. ICI targets mainly include PD-1, PD-L1, and cytotoxic T lymphocyte antigen-4 (CTLA-4) [20]. The emergence of ICI has brought new research directions to researchers, and we look forward to better development of immune checkpoint inhibitors in the future.

In our study, by using publicly data from the Liver Hepatocellular Carcinoma database of The Cancer Genome Atlas (TCGA-LIHC), we identified 434 significantly differently expressed oxidative stress related genes (DEOSGs) in HCC. To discover the underlying mechanisms, protein-protein interaction (PPI) network and enrichment analysis were subsequently employed on DEOSGs [21,22,23]. Subsequently, the WGCNA analysis of 434 DEOSGs identified 257 hub genes related to HCC, and 146 prognostic DEOSGs were then recognized using univariate cox analysis. Furthermore, the prognostic risk model was determined by using LASSO analysis and multivariate cox regression analysis [24]. In addition, the correlation of the prognostic model and clinical characteristics, immune microenvironment, and immunotherapy response of HCC was explored in TCGA-LIHC cohort. Based on our information, this is the novel prognostic OS-related risk model for HCC patients leading to improved prediction of HCC prognosis and personalized treatment management for HCC immunotherapy.

## 2. Material and Methods

### 2.1. Data Processing and DEOSGs Identification

From the TCGA and ICGC databases, defined respectively as the TCGA-LIHC cohort and ICGC cohort, transcriptomic, genomic mutation, and clinical messages were extracted. From the GeneCards Database, a list of 1399 OS genes with a relevance score ≥7 was obtained. The limma package in R was utilized to compare OS genes expression between normal tissues and HCC tissues for the purpose of screening DEOSGs, as the following standard: |log2 fold change| (|log2FC|) > 1 and adjusted *p* value < 0.05. This yielded 434 DEOSGs for subsequent analysis.

### 2.2. PPI Network Construction

A significant association of the PPI network in DEOSGs was created by employing the STRING database. We also employed Cytoscape software (version 3.7.2, Seattle, WA, USA) for the visual exploration of interaction networks. Molecular Complex Detection (MCODE) was operated as a Cytoscape plugin to identify critical PPI network modules [23].

### 2.3. Functional Enrichment Analysis

We conducted Kyoto Encyclopedia of Genes and Genomes (KEGG) and Gene Ontology (GO) analyses for these DEOSGs by employing the clusterProfiler package to determine their main biological and underlying mechanisms [21,22]. Adjusted *p* value < 0.05 was regarded as statistically significance.

### 2.4. Identification of HCC-Related Hub Genes by WGGCNA

The “WGCNA” R package was performed to build a co-expression network based on 434 DEOSGs in the TCGA database [25,26]. Using hierarchical clustering, 434 DEOSGs were analyzed among normal and tumor tissues. Next, a suitable soft thresholding parameter (β) was applied to emphasize gene connections and penalize weak connections. Afterward, we converted the adjacencies into topological overlap matrices (TOMs). By using the TOM-based dissimilarity assessment, the DEOSG dendrogram was clustered with a minimum module size of 25 and the dissimilarity between modules was measured. Furthermore, the modules most relevant to tumor tissues were found by two parameters (gene significance (GS) and module eigengenes (MEs)). In functional terms, hub genes are nodes within modules that have a high degree of interconnection [27]. This study defined genes in the significant module as HCC-related hub genes.

### 2.5. Establish and Validation of Prognostic Risk Model

In the TCGA-LIHC cohort, we carried out the univariate cox analysis among hub genes to determine DEOSGs closely related to overall survival (OS) via the “survival” R package, and genes were recognized as prognostic DEOSGs with *p* < 0.05. Afterwards, the LASSO regression analysis was conducted using the glmnet package in R software [24]. Finally, the prognostic risk model was created by multivariate cox analysis. For evaluating HCC risk scores, the following formula was used: risk score = ∑ni = vi × ci (where vi means gene’s expression and ci means gene’s corresponding coefficient). In the ICGC cohort, the risk model was confirmed. Each patient’s risk score was determined using the formula above. With the median value as the cut-off point, patients were stratified into high and low risk groups. The prognostic value of the model was assessed by Kaplan–Meier survival curves (K–M curves). Specificity and sensitivity of our risk model was validated by creating receiver operating characteristic (ROC) curves using the survivalROC and timeROC R packages.

### 2.6. Construction of Nomogram

On the TCGA cohort, univariate and multivariate analyses were conducted to assess if the risk model could be regarded as an independent risk factor. An evaluation of the clinical usefulness of the model at different risk thresholds was also conducted using the decision curve analysis (DCA) function in the “ggDCA” R package [28]. Comprised of gender, pathological grade, age, tumor stage, and risk group, a prognostic nomogram was established. A concordance index (C-index) and calibration curve were constructed for verifying the predicting discriminative value and accuracy of the nomogram. The construction of the nomogram, C-index, and calibration plots were completed using the “RMS” R package [29,30].

### 2.7. Calculation of Tumor Mutation Burden

Tumor mutational burden (TMB) is measured by the number of mutations per million bases in a tumor sample, including nonsense mutations, frameshift mutations, missense mutations, etc. Based on 38 million human exons, we calculated the TMB values using Perl scripts. In order to visualize the relationship among the TMB and risk group, we generated waterfall plots with the “maftools” package in R [31].

### 2.8. Investigation of HCC Immunity

We analyzed immune subtype proportions between the two risk groups based on UCSC-Xena database (https://xenabrowser.net//datapages//, accessed on 25 August 2022) using the “RColorBrewer” package [32]. Furthermore, to further confirm the connection between the risk and HCC immunity, the immune cell infiltration levels were quantified with ssGSEA. By using the Wilcoxon test of two samples, the levels of immune cells and the function of the two groups were compared. A comparison was also made between different groups regarding the expression of immune checkpoint genes and HLA (human leukocyte antigen) genes.

To predict immunotherapy response, we obtained immunophenoscores (IPS) for HCC patients using the Cancer Immunome Atlas (TCIA, https://tcia.at/, accessed on 25 August 2022) website [33]. Wilcoxon rank-sum tests were used to investigate the relationship between IPS and risk model. An increase in IPS score reflects higher immune reactivity, and a lower IPS score indicates lower immune reactivity.

### 2.9. Expression and Genetic Alteration Analyses of DEOSGs of Risk Model in HCC

TCGA-LIHC dataset was utilized to analyze the mRNA expression of five DEOSGs of risk model in HCC, and the Human Protein Atlas (HPA) (https://www.proteinatlas.org/, accessed on 25 August 2022) was applied to determine protein expression of five DEOSGs [34]. Prognostic evaluation of five DEOSGs was completed using the Kaplan–Meier Plotter website (https://kmplot.com/analysis/, accessed on 25 August 2022). In addition, to detect the copy-number alterations and mutation of above genes, the online database cBioPortal was employed [35].

### 2.10. Cell Culture

The human normal liver cell line, THLE-2, was obtained from Shanghai Academy of Life Science (Shanghai, China). Human hepatoblastoma cell line HepG2 (catalog No. ZQ0022), human HCC cell lines, (HCCLM3 (catalog No. ZQ0023), Hep3B (catalog No. ZQ0024), and Huh7 (catalog No. ZQ0025) were purchased from Zhong Qiao Xin Zhou Biotechnology (Shanghai, China). All cell lines were inoculated into culture dishes and added to Dulbecco’s Modified Eagle’s Medium (DMEM) containing 10% fetal bovine serum (FBS) to maintain growth. The cell culture medium was also supplemented with 1% penicillin/streptomycin. The growth environment temperature was maintained at 37 °C with 5% CO_2_. The culture dishes were purchased from Guangzhou Jet Biofiltration (Guangzhou, China). FBS, and penicillin/streptomycin were purchased from BI (BI, Ridgefield, CT, USA).

### 2.11. RNA Extraction and Quantitative Reverse Transcription PCR

Using the total RNA isolation kit (Foregene, Chengdu, China) and the PrimeScript RT kit (Takara Biomedical Technology, Beijing, China), RNA was extracted from the cells and reverse transcribed into cDNA. TB Green Premix Ex TaqII (Takara Biomedical Technology, Beijing, China) was used for real-time PCR. The relative expression was compared with the expression of GAPDH, and was calculated by 2−ΔΔCt. The qPCR primers are listed in Appendix A. [36]

### 2.12. Chemotherapeutic Drug Analysis in Different Groups

Since there are no biomarkers that predict HCC chemotherapy drug sensitivity; through 10-fold cross-validation, the half-maximal inhibitory concentrations (IC50) of several chemotherapy drugs among two risk groups were calculated and compared. The calculation process was carried out by ridge regression using the “pRRophetic” R package [37], which can predict tumor chemotherapy response based on gene expression levels. A comparison of IC50 among the two groups was processed by the Wilcoxon signed-rank test.

### 2.13. Statistical Analysis

Perl language and R language were applied to carry out all statistical analyses. The differences in measurement data were compared by Student’s *t*-test. Using the log-rank test, the Kaplan–Meier curve was established for identifying survival of HCC patients. To determine independent predictors of HCC survival, Cox regression analyses were further executed. The Chi-squared test was used to compare data from categorical variables between two groups. All statistical analysis were executed with R software (version 4.1.0, https://www.r-project.org/, accessed on 25 August 2022). Two-tailed significance tests were performed, with *p* < 0.05 regarded as statistically significant. * *p* < 0.05; ** *p* < 0.01; *** *p* < 0.001; **** *p* < 0.0001.

## 3. Results

### 3.1. Analysis of DEOSGs in HCC Samples

In the TCGA-LIHC cohort, expressions of 1399 OS genes were compared between 50 normal and 374 HCC samples. Of these, 434 oxidative stress related genes (OSGs) were identified as DEOSGs in HCC, comprising 62 downregulated and 372 upregulated genes (Appendix A, *p* < 0.05 and |log2FC| ≥ 1). As shown in Figure 1A,B, a hierarchical cluster heatmap and volcano plot were employed to visualize the DEOSGs.

To explore the interrelationship of DEOSGs, by employing the STRING dataset and Cytoscape software, we created a PPI network with 416 vertices and 6280 links (Figure 1C). As shown in Figure 1D, the MCODE plugin was further used, and the top 3 substantial modules (with 45 vertices and 498 links, 50 vertices and 466 links, and 40 vertices and 132 links) were identified as the potential key modules in the network.

Figure 1E–H shows the major 30 enriched pathways recognized by KEGG pathway analysis, including lipid and atherosclerosis, cellular senescence, MAPK signaling pathway, hepatitis B, and other KEGG pathways. As shown in Figure 1I–L, the leading 10 enriched GO terms for DEOSGs included oxidative stress, aging, and response to hypoxia. The functional enrichment analysis discovered that the DEOSGs were connected to cellular senescence, MAPK pathway, aging, and cancer.

### 3.2. Identification of HCC-Related Hub DEOSGs for by WGCNA

After that, WGCNA of 50 normal and 374 HCC samples from the TCGA database was performed based on the expression of 434 DEOSGs (Figure 2A). A soft threshold of 4 was chosen for the scale-free networks (Figure 2B,C); and three co-expressed modules were identified (Figure 2D). Each module was colored differently in order to determine the one most closely related to HCC. In particular, the turquoise module was chosen (Cor = 0.23, *p* < 0.0001) (Figure 2E). In total, 257 genes in turquoise module were identified as hub DEOSGs of HCC (Figure 2F).

### 3.3. Construction of a Prognostic Risk Model of DEOSGs

A univariate cox regression analysis was applied on 257 hub DEOSGs in order to identify prognosis-associated DEOSGs, and 146 DEOSGs with *p* < 0.05 were recognized as HCC prognostic DEOSGs (Appendix A). Furthermore, the LASSO algorithm was used for calculating the collinearity and following refinement (Figure 3A,B), and 5 DEOSGs, including lysyl oxidase (*LOX*), cytochrome P450 family 2 subfamily C member 9 (*CYP2C9*), eukaryotic translation initiation factor 2B subunit delta (*EIF2B4*)**,** enhancer of zeste 2 polycomb repressive complex 2 subunit (*EZH2*), and sulfiredoxin 1 (*SRXN1*) were selected to form a prognostic risk model (Figure 3C, Appendix A). The prognostic risk score were calculated as follows: risk score = 0.246 × *LOX* + (−0.083) × *CYP2C9* + 0.551 × *EIF2B4* + 0.356 × *EZH2* + 0.353 × *SRXN1*.

HCC patients were categorized into low and high risk groups based on their median risk scores in both the TCGA-LIHC and ICGC cohorts. In both the TCGA-LIHC cohort (Figure 3D) and ICGC cohort (Figure 3H) (*p* < 0.001), HCC patients with higher scores had worse overall survival. Notably, with an increased risk score, the death events were significantly increased (Figure 3E,I). Besides, time-dependent ROC curves showed the prognostic risk model to be quite reliable with the area under the ROC (AUC) of 0.790 in the TCGA-LIHC cohort (Figure 3F) and 0.743 in the ICGC cohort (Figure 3J) at 1 year. More interesting, in the TCGA-LIHC cohort, compared to other clinical characteristics, such as age, gender, pathological grade, tumor stage, T, M, and N, AUC over 1 year demonstrated that the model had an higher prognostic accuracy (Figure 3G). Moreover, the model showed good predictive ability in the ICGC cohort (Figure 3K). Taken together, this prognostic risk model had sufficient specificity and sensitivity.

### 3.4. Independent Prognostic Value of the Model and Establish of the Nomogram

Moreover, to determine the independence the risk model for HCC patients, independently prognostic analysis was achieved using univariate and multivariate cox regression analysis. In the TCGA-LIHC cohort (Figure 4A,B) and the ICGC cohort (Figure 4C,D), the risk model maybe an independently prognostic factor of HCC patients (*p* < 0.001). In the TCGA-LIHC cohort, as shown by the DCA curve (Figure 4E), the model outperformed other clinicopathological characteristics.

The survival probabilities for HCC at 1, 3, and 5 years were estimated using a nomogram plot containing the risk scores and clinical characteristics of the TCGA-LIHC cohort (Figure 4F). Moreover, the C-index of the risk score was higher than other clinicopathological characteristics (Figure 4G). The calibration plots indicated great consistency between prediction and actual risk in the TCGA-LIHC cohort at 1, 3, and 5 years (Figure 4H). Overall, the prognostic risk model showed good prognostic value in HCC samples.

### 3.5. Comparison of Risk Model and Other Models

We compared our risk model with three established prognosis-related risk models, namely 4-gene signature (Peng) [38], 8-gene signature (Qu) [39], and 8-gene signature (Cao) [40]. Based on the corresponding genes and coefficients, risk scores were calculated for each HCC patient using the three models. The median risk scores were used to classify HCC patients into high and low risk groups. The three models showed different prognoses of HCC in the high and low risk groups (Figure 5A–C, *p* < 0.001). However, our model had higher 1, 3, and 5-year AUC values (Figure 5D–F) and a higher C-index (Figure 5G) than these models, highlighting that our model has more reasonable and effective results.

### 3.6. Connections between Prognostic Risk Model and Clinical Characteristics of HCC Patients

In the TCGA–LIHC cohort, the distribution of clinical characteristics, including survival state, pathological grade, and tumor stage, among the two risk groups was significantly different (Figure 6A, *p* < 0.05). Additionally, HCC patients with higher risk scores had the worst pathological grades and advanced tumor stages (Figure 6B–D) of HCC patients.

According to the different clinical parameters, we further divided patients into several subgroups (Figure 6E–N). In this study, the clinical stratifications contained gender (female vs. male) (Figure 6E,F), pathological grade (G1 vs. G2 vs. G3–4) (Figure 6G–I), age (≥60 vs. <60) (Figure 6J,K), and tumor stage (I vs. II vs. III–IV) (Figure 6L–N). According to the K–M curves, for all of these parameters, with the exception for stage II, high-risk patients had worse outcomes than low-risk patients.

### 3.7. Functional Analyses Based on the Risk Model

Gene function analyses among the two groups were conducted by 2195 DEGs identified between the subgroups (*p* < 0.05 |log2FC| ≥ 1). GO enrichment analysis revealed that DEGs tended to be enriched in “nuclear division”, “positive regulation of cell activation”, “B cell activation”, and “mitotic nuclear division” in the biological process (Figure 7A–C). DEGs were primarily concentrated in the PI3K-Akt signaling pathway, cell cycle, cytokine-cytokine receptor interaction, ECM-receptor interaction, and Fc gamma R-mediated phagocytosis, according to the KEGG pathway analysis (Figure 7D–F). The functional annotations from GO and KEGG were further validated and complemented by gene set enrichment analysis (GSEA). A GO enrichment analysis of the high risk group identified the following biological processes as the most enriched: activation of immune response, B cell activation, adaptive immune response, and B cell mediated immunity (Figure 7G). In contrast, drug catabolic process, high density lipoprotein particle, and aromatase activity were the majority concentrated pathways in the low risk group (Figure 7H). Additionally, based on KEGG analysis, the high risk group’s enriched pathways included cell adhesion molecules (CAMs), ECM-receptor interaction, and cytokine-cytokine receptor interaction (Figure 7I), while fatty acid metabolism, butanoate metabolism, and primary bile acid biosynthesis were the most concentrated pathways in low risk group (Figure 7J).

### 3.8. Analysis of Tumor Mutation Burden

TMB, or non-synonymous variation, plays a significant role in immune cell infiltration [41]. As shown in Figure 8A, the waterfall plot indicated the mutation rate of 146 prognostic DEOSGs in the TCGA cohort. According to Spearman’s correlation analysis, the risk score and mRNAsi score are positively correlated (Figure 8B). Since TMB may have a significant role in clinical practice, the relationship of TMB and the model was investigated. TMB was higher in the high risk group, and was positively correlated with risk score (Figure 8C,D). Furthermore, the TMB and risk score combined to influence survival outcomes in HCC patients (Figure 8E). Using waterfall plots, different risk groups were shown to have different mutation characteristics (Figure 8F,G). In the high risk group, mutation frequency was higher, in line with the TMB score. Increased mutation frequencies of *TP53*, *CTNNB1*, *TTN*, and *MUC16* were found in the high risk group, but the *ALB* mutation frequency was noticeably greater in the low risk group. The majority of the mutations were missense mutations and frameshift deletions.

### 3.9. Associations between Prognostic Risk Model and HCC Immunity

According to Figure 9A, the proportion of HCC immune subtypes varied between risk groups. In Figure 9B,C, differences between two risk groups are shown in terms of immune cell and immune function scores. Furthermore, two groups were compared regarding HLA gene expression and immune checkpoint gene expression. The results showed that high risk patients had higher expressions of the majority of HLA genes and immune checkpoint genes (Figure 9D,E), indicating an exhausted immune microenvironment in patients with higher risk scores. Furthermore, Appendix A revealed the detailed information of several immune checkpoint genes’ expression in the high and low risk groups: patients in the high risk group had higher expressions of *PD-1* (*p* < 0.001) (Appendix A), *CTLA4* (*p* < 0.001) (Appendix A), *TIM3* (*p* < 0.001) (Appendix A), *TNFSF4* (*p* < 0.001) (Appendix A), *TNFSF9* (*p* < 0.001) (Appendix A), *TNFSF18* (*p* < 0.001) (Appendix A), *TNFRSF4* (*p* < 0.001) (Appendix A), *TNFRSF9* (*p* < 0.001) (Appendix A), *TNFRSF18* (*p* < 0.001) (Appendix A), *CD276* (*p* < 0.001) (Appendix A), *VTCN1* (*p* < 0.001) (Appendix A), and *TIGIT* (*p* < 0.001) (Appendix A). A high risk score in the risk model was significantly positively associated with *PD-1* (R = 0.27, *p* < 0.001), *CTLA4* (R = 0.33, *p* < 0.001), *TIM3* (R = 0.33, *p* < 0.001), *TNFSF4* (R = 0.3, *p* < 0.001), *TNFSF9* (R = 0.32, *p* < 0.001), *TNFSF18* (R = 0.25, *p* < 0.001), *TNFRSF4* (R = 0.38, *p* < 0.001), *TNFRSF9* (R = 0.37, *p* < 0.001), *TNFRSF18* (R = 0.3, *p* < 0.001), *CD276* (R = 0.45, *p* < 0.001), *VTCN1* (R = 0.26, *p* < 0.001), and *TIGIT* (R = 0.2, *p* < 0.001). In addition, Appendix A revealed that *SRXN1*, *LOX*, *EZH2*, and *EIF2B4* were positively correlated with immune checkpoint genes, while *CYP2C9* were negatively correlated with immune checkpoint genes.

Figure 9F revealed the correlation between five genes in models and immune cells infiltration. Further evaluation was conducted to determine if the model could predict HCC patients’ response to immunotherapy. As shown in Figure 9G–J, the value of *CTLA4^−^ PD1^−^*-IPS and *CTLA4^+^ PD1^+^*-IPS in the low risk group was higher than in the high risk group (*p* < 0.05), indicating that HCC patients in the low risk group showed significant therapeutic effects with anti-*CTLA4* therapy and anti-*PD1* therapy.

### 3.10. Expression of DEOSGs in Prognostic Risk Model in HCC

Furthermore, we explored the expression of five DEOSGs as a prognostic risk model for HCC. In the TCGA database, the mRNA level of *LOX*, *EIF2B4*, *EZH2*, and *SRXN1* were significantly upregulated in HCC samples, while *CYP2C9* was significantly downregulated compared with normal liver tissues (Figure 10A). The K–M curves revealed that the high expression of *LOX*, *EIF2B4*, *EZH2*, and *SRXN1* presented a poor OS. In contrast, high *CYP2C9* expression predicted better OS (Figure 10B). Similarly, from the HPA dataset, the protein expression pattern of these five genes demonstrated a consistent trend in mRNA expression (Figure 10C); unfortunately, there is no *LOX* and *SRXN1* information in this database. As shown in Figure 10D, the frequencies of gene alterations, containing amplifications, deep deletions, missense mutations, and truncating mutation for these five genes ranged from 0.2% to 1.7% in the cBioPortal database.

### 3.11. RT-qPCR Validation of Five Genes in HCC Cell Lines

We used RT-qPCR to further verify the expressions of model genes (*CYP2C9*, *EIF2B4*, *EZH2*, *SRXN1*, and *LOX*) in the HCC cell lines. The results showed that the expressions of *EIF2B4*, *EZH2*, *SRXN1*, and *LOX* were upregulated in most HCC cell lines, and the expressions of *CYP2C9* was downregulated, which was consistent with gene expressions in the TCGA database (Figure 10E–I).

### 3.12. Drug Sensitivity Analysis of the Model

To measure whether the model could be applied to personalized treatment of HCC, we compared the IC50 values of several chemotherapy drugs between two groups by the Wilcoxon signed-rank test. A low IC50 indicates a sensitive response. Interestingly, a curious finding was that compared to the low risk group, patients with high risk had lower IC50 values of sorafenib (*p* < 0.0001), sunitinib (*p* < 0.0001), dasatinib (*p* < 0.0001) gemcitabine (*p* < 0.0001), rapamycin (*p* < 0.0001), roscovitine (*p* < 0.0001), paclitaxel (*p* < 0.0001),mitomycin C (*p* < 0.0001), and bleomycin C (*p* < 0.0001), suggesting that the above chemotherapies may benefit patients with high risk (Figure 11).

## 4. Discussion

HCC, as a malignant neoplasm, is the third common cause of cancer death, globally accounting for almost 75–85% of primary liver cancer deaths. Although targeted molecular therapies and immunotherapy have demonstrated encouraging treatment benefits for HCC patients recently, and atezolizumab and bevacizumab are used as first-line treatment for advanced HCC [42], the overall clinical prognosis of HCC patients are far from satisfactory as a consequence of drug resistance [1,4]. Many studies were performed to explore effective models to precisely predict the prognosis and response to treatment of HCC patients, in the hope of providing more evidence for precision therapy [43,44,45]. Therefore, it is imperative to detect novel validated prognostic biomarkers and construct new models to predict the precise treatment for HCC patients.

An increase in ROS and RNS causes oxidative stress [46], which plays a key factor in the progression of liver carcinogenesis [47]. The presence of elevated ROS levels causes damage to DNA, RNA, lipids, and proteins. ROS imbalance promotes cellular proliferation, apoptosis evasion, angiogenesis, invasion, and metastasis [48]. When ROS are highly expressed, immature myeloid cells (IMC) produce MDSCs that suppress the immune system. Therefore, the clinical significance, biological role, and immune infiltration of OS genes can be extensive analyzed to provide a new direction in the research and treatment on HCC.

In our study, based on the TCGA database, 434 DEOSGs were determined. Of these, we regarded 257 genes in the turquoise module identified by WGCNA as HCC-related hub genes. Then, 146 prognostic DEOSGs were found by univariate cox analysis, and five DEOSGs (*LOX*, *CYP2C9*, *EIF2B4*, *EZH2*, and *SRXN1*) were selected to develop a risk model of HCC patients by LASSO and multivariate cox analysis.

Afterwards, this prognostic model had a superior forecast accuracy than the usual clinical characteristics, including pathological grade, tumor stage, age, and gender, as indicated by the ROC curve. More importantly, we revealed that the risk model was remarkably connected with advanced tumor stage, and worse pathological grade. Taken together, our results demonstrate this DEOSGs-related risk model can improve the prediction of prognosis and supply a novel perspective for assessing the survival in HCC patients.

We observed that the low risk group was mainly enriched in metabolism-related pathways, while the high risk group was primarily related to immune pathways. Furthermore, we found that TMB and the risk score can effectively distinguish the prognosis of HCC patients. Additionally, we displayed the different gene mutations frequency between subgroups.

More interestingly, C1 (wound healing) and C2 (IFN-g dominant) subtypes were more prevalent in the high risk group, whereas C3 subtypes (inflammatory) were less prevalent. The expression of immune checkpoint genes enable tumor cells to evade immune surveillance, resulting in immunosuppression in the TME. The expression of almost all immune checkpoint genes were upregulated in patients with high risk scores, and correlation analyses showed significant positive association between checkpoint genes and risk score, as well as the five OS genes comprised in the risk model, suggesting that patients with higher risk scores have severer exhausted immune microenvironments. In addition, the low risk group patients respond better to anti-*PD-1* and anti-*CTLA4* therapies, showing that immunotherapy can benefit HCC patients in the low risk group.

The TCGA and HPA datasets showed that expression of *LOX*, *EIF2B4*, *EZH2*, and *SRXN1* were significantly upregulated in HCC tissues, while *CYP2C9* was significantly downregulated. The K–M curves of these five genes showed that the upregulation of *LOX*, *EIF2B4*, *EZH2*, and *SRXN1* presented poor OS. However, a higher expression of *CYP2C9* predicted better OS. Umezaki et al. discovered that high expression of *LOX* is associated with EMT markers, early recurrence, and poor survival in HCC patients [49]. Research has also reported that decreased levels of *CYP2C9* in HCC tissues [50,51], indicating the involvement of *CYP2C9* in detoxification, may play an important role in the initiation of HCC. There is evidence that *EZH2* is elevated in HCC and promotes proliferation, migration, and invasion of HCC cells, acted as a negative prognostic biomarker associated with immunosuppression, and is associated with sorafenib resistance in HCC [52,53,54,55]. *SRXN1*, an antioxidant molecule, was significantly upregulated in HCC and its increasement might lessen *BTG2* expression, resulting in an increase in HCC cell proliferation. These findings are broadly consistent with the expression patterns we detected in this study. Notably, the function of *EIF2B4* in HCC have not been reported yet and need to be further investigated.

It should be noted that HCC is relatively insensitive to chemotherapeutic drugs because of drug resistance mechanisms and heterogeneity, so that chemotherapeutics have limited effect. Several chemotherapy drugs showed different responses in the two groups, suggesting that the model could also help in selecting chemotherapy drugs for HCC patients.

Although we discovered a risk model composed of five DEOSGs that proved to be related to immunosuppression, there are limitations to this work. Further studies are needed to validate its reliable prognostic value in clinical practice. Meanwhile, the study was a retrospective analysis and conducted by bioinformatics analysis, which was not powerful enough and requires further prospective research and experimental verification. In addition, a number of laboratory experiments, including in vivo and in vitro studies, need to be performed to determine the underlying mechanism of these genes in HCC progression.

In our study, we constructed a prognosis-related oxidative stress model with strong predictive power for HCC patients. The phenotypes of HCC patients can be quantified and individualized using the risk score calculated by the model. The model may have important implications for the selection of ICIs and chemotherapy for HCC. Taken together, our model has great potential as a novel biomarker of HCC. As far as we know, this is the first prognostic OS-related risk model for HCC patients leading to improving prediction of HCC prognosis and providing new insights for HCC immunotherapy.

## Figures and Tables

**Figure 1 biomedicines-11-00695-f001:**
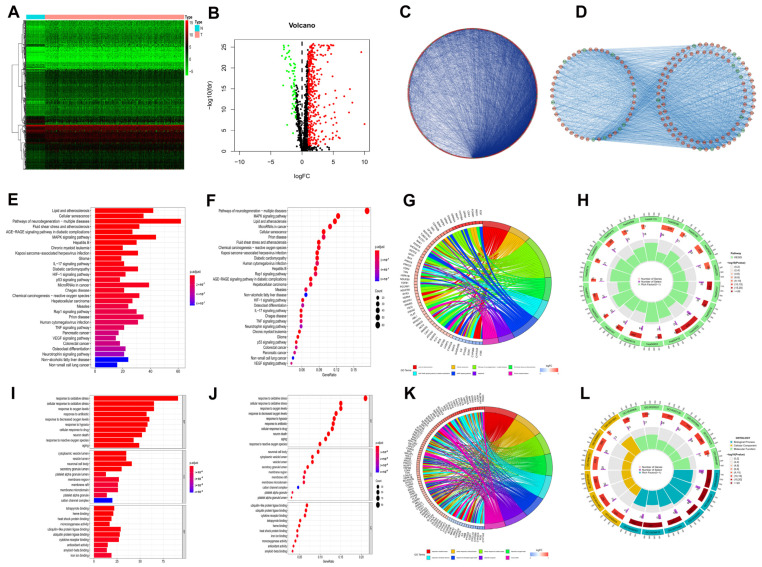
DEOSGs analyses in HCC. (**A**) Heatmap shows downregulated (green) and upregulated (red) genes between HCC and normal liver tissues in the TCGA−LIHC dataset. (**B**) The volcano plot for the 434 DEOSGs (green and red plots represent downregulated and upregulated genes, respectively, black plots represent non−differentially expressed genes.) (**C**) PPI Network of DEOSGs. (**D**) Top 3 significant modules form PPI network: red circles, upregulated genes; green circles, downregulated genes. Barplot (**E**), bubble plot (**F**), chord diagram (**G**), and circle plot (**H**) of KEGG analyses of DEOSGs. Barplot (**I**), bubble plot (**J**), chord diagram (**K**), and circle plot (**L**) of GO analyses of DEOSGs.

**Figure 2 biomedicines-11-00695-f002:**
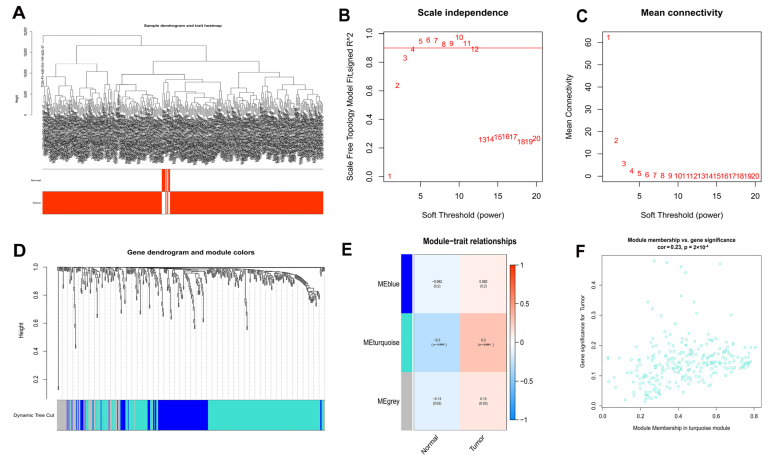
WGCNA analysis results of 434 DEOSGs. (**A**) Clustering dendrogram of HCC samples. (**B**,**C**) The scale–free fit index for soft–thresholding powers. (**D**) A dendrogram of the differentially expressed genes clustering based on different metrics. (**E**) A heatmap showing the correlation between the gene module and associated traits. (**F**) Scatter plot of module eigengenes in turquoise modules.

**Figure 3 biomedicines-11-00695-f003:**
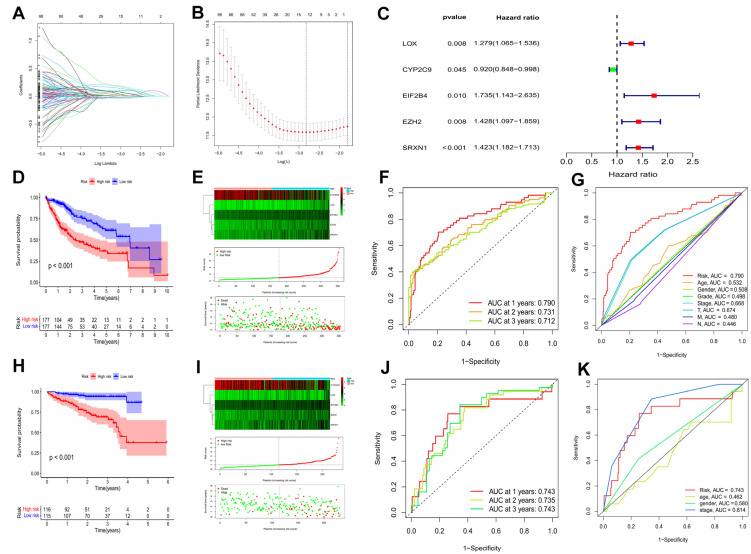
Identification of a prognostic risk model for HCC patients. (**A**) LASSO coefficient profiles of the 12 DEOSGs with non-zero coefficients determined by the optimal lambda. (**B**) Screening of optimal parameter (lambda) at which the vertical was drawn. (**C**) Multivariate cox regression of five candidate DEOSGs genes. Kaplan–Meier curve of HCC patients in the TCGA-LIHC cohort (**D**) (*p* < 0.001) and the ICGC cohort (**H**) (*p* < 0.001). The genes expression heatmap, risk score distribution, and survival status of the model in the TCGA-LIHC cohort (**E**) and the ICGC cohort (**I**). TimeROC curves in the TCGA-LIHC cohort (**F**) and the ICGC cohort (**J**). Clinical ROC analysis in the TCGA-LIHC cohort (**G**) and the ICGC cohort (**K**).

**Figure 4 biomedicines-11-00695-f004:**
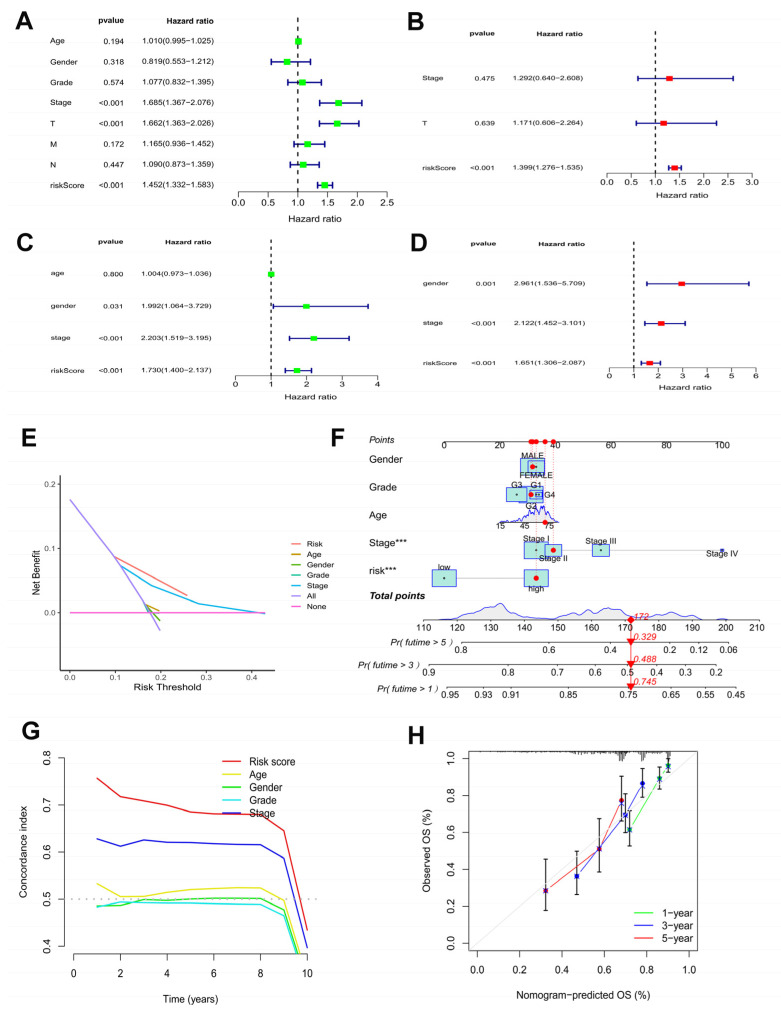
Efficacy estimate of prognostic model and nomogram. Univariate cox regression analysis of the model in the TCGA-LIHC cohort (**A**) and the ICGC cohort (**C**). Multivariate cox regression analysis of the model in the TCGA-LIHC cohort (**B**) and the ICGC cohort (**D**). (**E**) The decision curve analysis of the model in the TCGA-LIHC cohort. (**F**) The nomogram consists of risk score and other clinicopathological parameters. (**G**) The C-index of the model and other clinicopathological parameters. (**H**) Calibration curves of the nomogram. *** *p* < 0.001.

**Figure 5 biomedicines-11-00695-f005:**
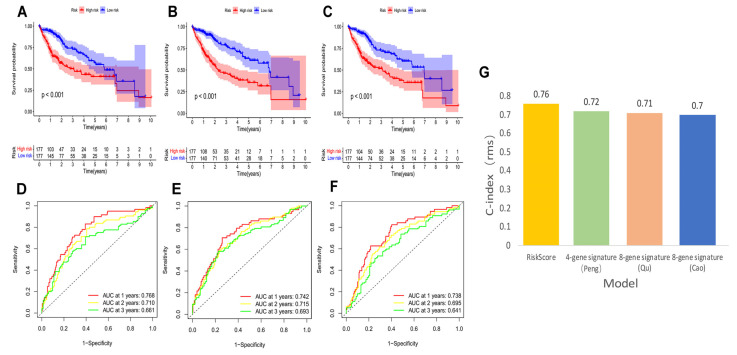
K–M curves, time-dependent ROC curves, and C-index of three other prognostic models in the TCGA–LIHC cohort. The HCC K–M curves in the high/low group of the 4-gene-signature (Peng) (**A**), the 8-gene-signature (Qu) (**B**), and the 8-gene-signature (Cao) (**C**); ROC curves of the 4-gene-signature (Peng) (**D**), the 8-gene-signature (Qu) (**E**), and the 8-gene-signature (Cao) (**F**); (**G**) comparison of C-index between our risk model and the three risk models.

**Figure 6 biomedicines-11-00695-f006:**
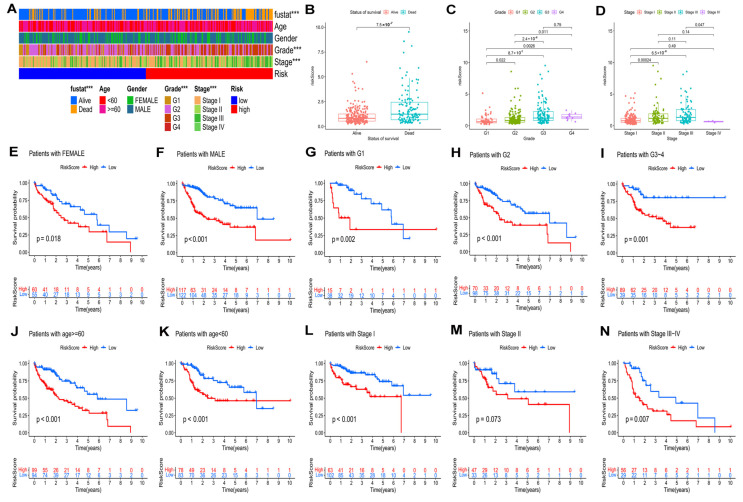
Relationship of risk model and clinical characteristics. (**A**) The heatmap of the model and clinical characteristics in the TCGA–LIHC cohort. Boxplot of risk score in HCC patients with different status of survival (**B**), pathological grades (**C**), and tumor stages (**D**). The K–M curves of HCC patients in different groups according to (**E**) female and (**F**) male; (**G**) Grade 1, (**H**) Grade 2, and (**I**) Grade 3–4; (**J**) age ≥ 60 and (**K**) age < 60; (**L**) stage I, (**M**) stage II, and (**N**) stage III–IV. *** *p* < 0.001.

**Figure 7 biomedicines-11-00695-f007:**
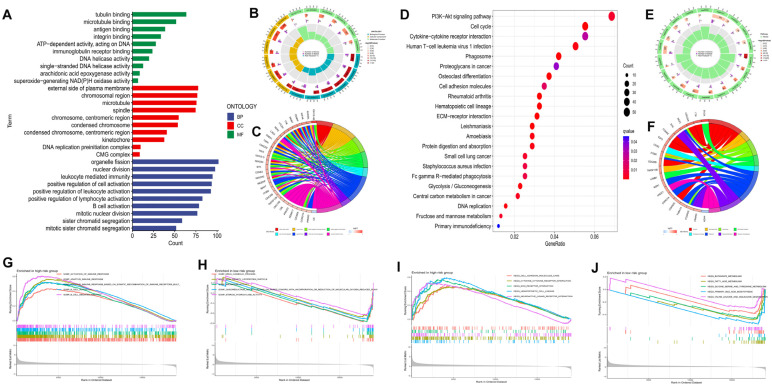
Functional analysis of DEGs among the high and low risk groups in the TCGA−LIHC cohort. The bar plot (**A**), circlize plot (**B**), and circular plot (**C**) of GO analysis the DEGs. The bubble plot (**D**), circlize plot (**E**), and circular plot (**F**) of KEGG analysis the DEGs. GO enrichment plots from GSEA in the high (**G**) and low risk group (**H**). KEGG enrichment plots from GSEA in the high (**I**) and low risk group (**J**).

**Figure 8 biomedicines-11-00695-f008:**
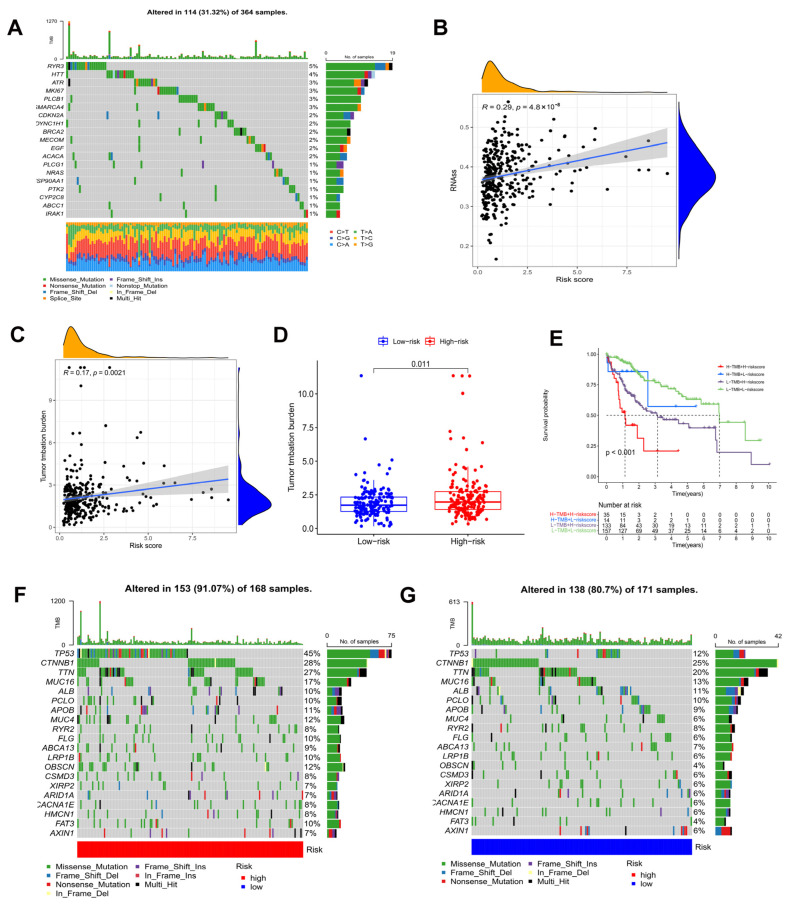
Tumor mutation analysis. (**A**) Genetic mutation frequency of 146 prognostic DEOSGs. (**B**) Correlation analyses between the model and mRNAsi scores (RNAss). (**C**) Correlation analyses between the model and TMB. (**D**) In the TCGA cohort, differences in TMB were observed between the two groups. (**E**) The Kaplan−Meier curve based on both TMB groups and the model for HCC patients. The waterfall plot showing the differences in somatic genomic mutation between high (**F**) and low risk groups (**G**).

**Figure 9 biomedicines-11-00695-f009:**
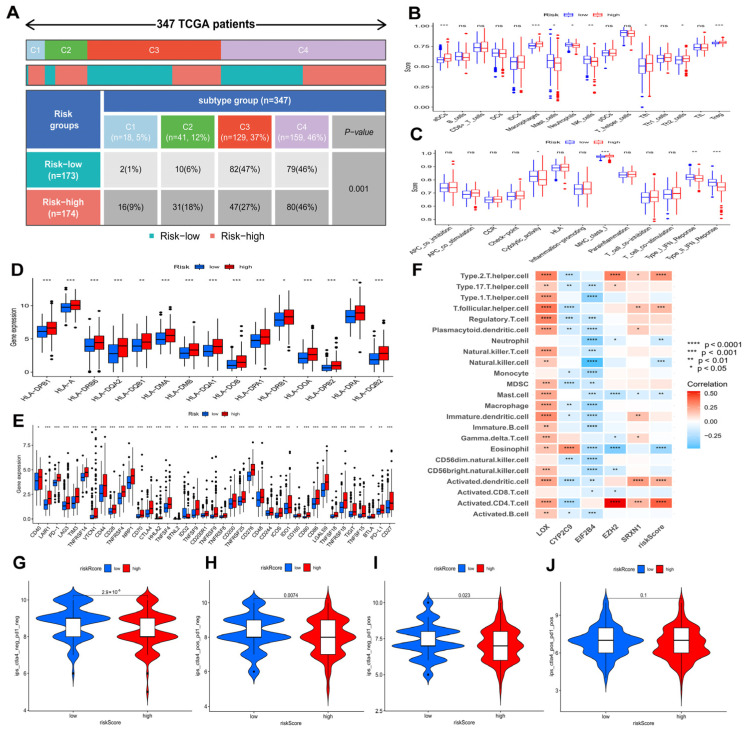
Associations of prognostic risk model and HCC immunity in the TCGA−LIHC cohort. (**A**) A comparison of immune subtypes between different risk groups. Immune cell (**B**) and immune function scores (**C**) of different risk groups. Different risk groups expressed different levels of HLA genes (**D**) and checkpoint genes (**E**). (**F**) The correlation of immune cells and model genes. Comparison of immunophenoscores (IPS) between two risk groups. (**G**) *CTLA4^−^ PD1^−^*, (**H**) *CTLA4^−^ PD1^+^*, (**I**) *CTLA4^+^ PD1^−^*, and (**J**) *CTLA4^+^ PD1^+^*. ns *p* > 0.05; * *p* < 0.05; ** *p* < 0.01; *** *p* < 0.001; **** *p* < 0.0001.

**Figure 10 biomedicines-11-00695-f010:**
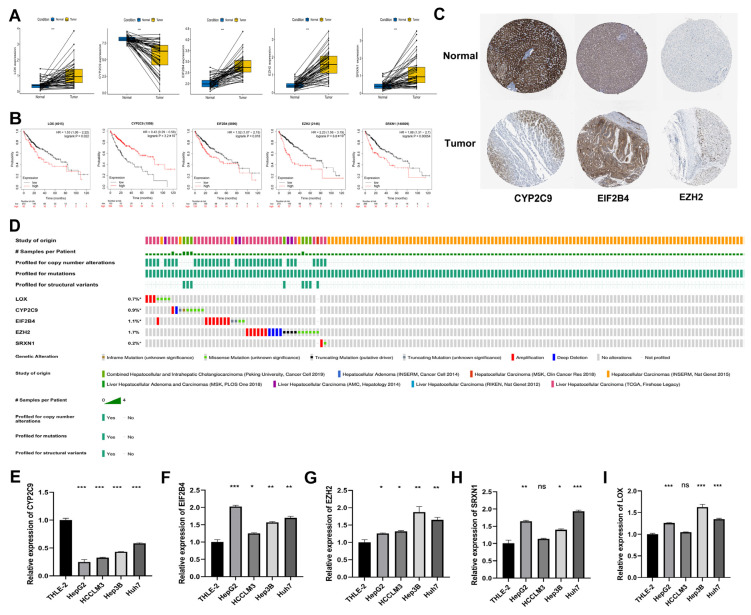
Expression, survival, and gene alteration analyses of model genes in HCC. (**A**) Transcription expression analysis of *LOX*, *CYP2C9*, *EIF2B4*, *EZH2*, and *SRXN1* in paired samples in the TCGA−LIHC dataset. (**B**) The K−M curves of *LOX*, *CYP2C9*, *EIF2B4*, *EZH2*, and *SRXN1* in the TCGA−LIHC dataset. (**C**) The protein level of *CYP2C9*, *EIF2B4*, and *EZH2* among normal liver tissues and HCC tissues in the Human Protein Atlas. (**D**) The frequencies of gene alterations of *LOX*, *CYP2C9*, *EIF2B4*, *EZH2*, and *SRXN1* in the TCGA−LIHC dataset. The mRNA expression levels of five genes in HCC cell lines. *CYP2C9* (**E**), *EIF2B4* (**F**), *EZH2* (**G**), *SRXN1* (**H**), and *LOX* (**I**). * *p* < 0.05; ** *p* < 0.01; *** *p* < 0.001.

**Figure 11 biomedicines-11-00695-f011:**
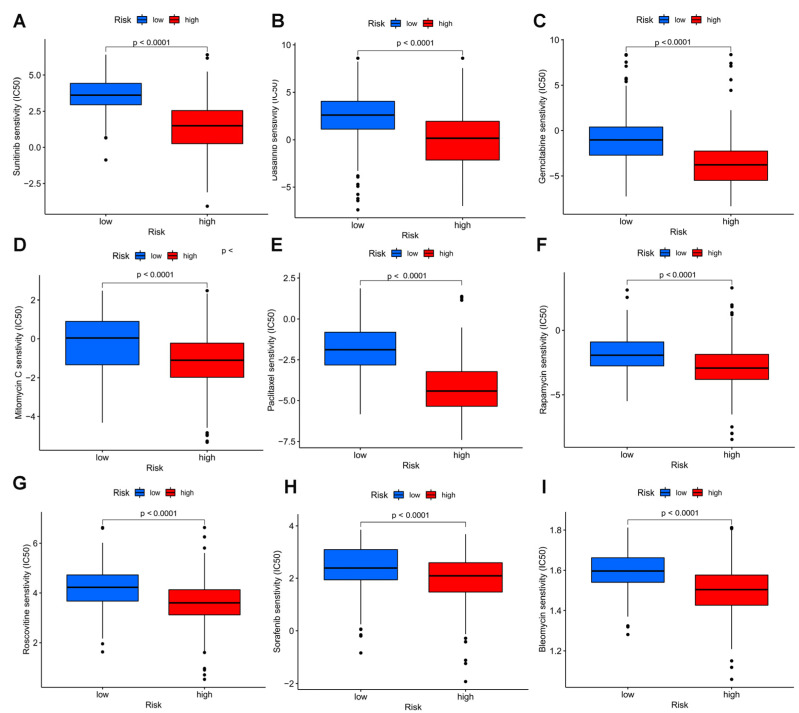
Drug sensitivity analyses among high and low risk groups. Estimated IC50 of sunitinib (**A**), dasatinib (**B**), gemcitabine (**C**), mitomycin C (**D**), paclitaxel (**E**), rapamycin (**F**), roscovitine (**G**), sorafenib (**H**), and bleomycin C (**I**) in the high risk group were lower than in the low risk group.

## Data Availability

The original contributions presented in the study are included in the article.

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
