# Peer review of "Identification of Prognosis-Related Oxidative Stress Model with Immunosuppression in HCC"

_biomedicines, 2023, doi:10.3390/biomedicines11030695_

Round 1

Reviewer 1 Report

Manuscript No biomedicines-2119372

Identification of Prognosis-related Oxidative Stress Genes with immunosuppression in HCC” for Biomedicines

Comments:

11. Introduction. Please also mention reactive forms of nitrogen, which also play an important role not only in physiology but also in pathological phenomena such as cancer.

22. Introduction. In the title, the authors indicate the relationship between oxidative stress and immunosuppression. In this chapter, please at least indicate the role of immunosuppression in HCC and a clear link between this process and OS.

33. Please explain all abbreviations used for the first time in the text.

44. Materials and methods. 2.10. Please indicate, if possible, the cell culture catalog numbers and briefly characterize them. The culture fluid definitely contained antibiotics. It should be supplemented. In addition, please fully describe the cell culture conditions.

55. Results. 3.11. Please specify clearly: IC50 lower than what? What has it been compared to? How much lower than in low risk patients?

66. Discussion. Please limit duplication of results in this chapter. In addition, I am asking for an unequivocal answer (according to the title of the work) about the connection between OS related genes and immunosuppression in HCC.

77. Figure 9. Please indicate in the figure caption what was compared with what. What the statistical significance asterisks refer to.

88. Please select the figures that are necessary for the main work and those that may be included in the supplementary materials.

Reviewer 2 Report

This interesting study focused on developing a prognostic oxidative stress-related model with a solid predictive potential as an immunosuppressive biomarker for HCC, improving treatment outcome prediction and providing new insights for HCC immunotherapy. The authors recognized and described well some of the limitations of the study and the need to validate further (in vitro and in vivo) the described increases in the four genes (LOX, EIF2B4, EZH2, and SRXN1) expression in the HCC samples used in the study, and decreased CYP2C9 gene expression. However, despite these limitations, the study provides new insights and knowledge into the overall undressing of changes in the oxidative stress-related genes of hepatocellular carcinomas (HCCs). 

Specific Comments:

1. The main question addressed by the research is to utilze a gene expression methodology for hepatocellular carcinoma (HCC) patients to establish a new oxidative stress-related prognostic model for predicting prognosis, exploring the immune microenvironment, and predicting what the immunotherapy response might be expected in response to therapy. 

2. I consider the topic both original and relevant in the field and provide some new insights into the oxidative stress genes of specific importance. However, the finding for the 4 genes that were increased and one decreased need to be further validated in vivo and in vitro separate studies because gene data is not sufficient to make solid conclusions for validity. So the paper is interesting and gives some directions for more work to be done but is not tell the whole story nor validate the findings at protein levels in different in vitro (PDX etc.) and in vivo human studies or animal models.

3. This study is focused on the different expressed oxidative stress-related genes(DEOSGs) between adjacent tissues and HCC tissues from the Cancer Genome Atlas (TCGA) were screened, then based on weighted gene coexpression network analysis (WGCNA), HCC-related hub genes were discovered. This is novel and different from other studies published wit HCC cells and tissues.

4. Results must be validated at protein leaves and confirmed in additional in vitro (PDX etc.) and in vivo human studies or animal models. However, it will be too much to ask for this work to be completed and added to this current manuscript so that it could be a separate future publication.

5. the data seems to be consistent with the conclusions, and both are well-aligned with the main question posed.

6. the references are appropriate.

7. The figure seems to be fine, maybe a little too small to read in printed PDF, but on the electronic copy, we could zoom in to make it easier for the eyes to read. So maybe some improvement in the formatting of the text and figures visually could be improved (that is a more editorial suggestion, not something that the authors could do alone).

Reviewer 3 Report

In this paper by Ren et al., the authors aim to develop an oxidative stress related prognostic model in order to predict immunotherapy response. Although the idea is interesting, several points need to be clarified.

Major points:

First of all, the quality of the provided figures is very unsatisfactory. Most of the figures are barely readable, which makes data reviewing very complicated. Therefore high quality and resolution figures should be provided to the reviewer.

Then, the legends of the figure are almost non-existent. There are no details helping to appreciate the provided data. Some of the panels are even not mentioned in the legends (see figure 1).

The rational for using OS genes as prediction markers is not clear. This should be better discussed given this is the central point of the manuscript.

The authors do not compare there OS-prognostic model with other relevant prognostic models for liver diseases and HCC already published. How do their model perform compare to this molecular models?

In the abstract the authors state that they compared adjacent tissues with HCC tissues, while in the results section (lines 191-192) they state that they compared 50 normal and 374 HCC tissues. Please explain and modify accordingly to what has been done.

The way the nomogram was establish is not clear and not detailed on the figures. Should be better explain because this is a central point of the MS.

Analysis of immunotherapy response in HCC patient is not clear (Fig 8 G-J). The response to anti-CTLA4 or anti-PD1 therapy does not appear clearly on the figure. Please give better explanation on how the analysis was performed.

What is the relevance of figure 9D? Please explain.

This is not clear why in figure 9E-H the mRNA expression in LO2 cell line is set to 1. Why the authors did not show LOX expression? The author state the genes are upregulated in HCC cell line. Upregulated compared to what? The author state in the methods LO2 cells are HCC cell line. In the result section they refer to the HL7702 cell line as normal liver cell line. LO2 and HL7702 are the same cell line. Moreover this has been clearly demonstrated this is not a liver cell line but a HeLa (cervix cancer) contaminated cell line that should therefore not be used as a relevant liver cell line.

It’s not clear how the drug sensitivity analysis was performed (figure 10). There are no details either in the methods or in the results sections. This should be better explain.

Overall, the connections between the different parts of the results are not clear. There is no obvious connecting thread. It’s not clear how the OS-related model could be applied in patients diagnosed with advanced HCC and how it would benefit these patients. There is an obvious lack of explanation on how the data were analyzed. The methods section does not give a lot of details. The figure legends do not give details at all. 

Minor points:

The authors should add a reference on the combination of atezolizumab + bevacizumab which is now an approved first line therapy for HCC.

The authors should provide references for the following statement: “many studies were performed to explore more effective models to precisely predict the prognosis and response to treatment of HCC patients for providing more evidence for precision therapy”.

English language has to be revised.

Some typos should be corrected.

Round 2

Reviewer 1 Report

Thanks to the authors for their work in making all the corrections. The manuscript, in my opinion, is sufficiently improved.

Author Response

Dear Reviewer,

Thank you for your recognition of our work,and we appreciate your very insightful comments.

Sincerely,

Xinhui Liu, M.D., Ph.D.

Integrated hospital of traditional Chinese medicine

Southern Medical University

Guangzhou, Guangdong, China, 510315

Reviewer 3 Report

The manuscript has been globally improved.

However, addressing the 2 following points would greatly reinforce the conclusion of the authors:

-        Figure 10: The authors should compare gene expression between HCC/hepatoblastoma cell lines and primary human hepatocytes. The LO2 cell line can definitely not be considered a “normal epithelial cell lline” given it’s “cell line” nature and the fact it’s contaminated with cervix cancer HeLa cell line. The comparison is there not appropriated. The wide usage of this cell line in several publications does not make it more valuable for liver studies.

-        Figure 11: Except sorafenib, the choice of the investigated chemotherapeutic drugs is questionable. Would be best to include approved first and second line therapies – Lenvatinib, regorafenib, cabozantinib, as well as ICI-including therapies - -atezolizumab + bevacizumab, nivolumab +/- ipilimumab. Given the prediction to immunotherapy response of the established model, it would make more sense.

Round 3

Reviewer 3 Report

The paper has been improved, and the authors answered my questions